# An Empirical Investigation of Implicit and Explicit Knowledge-Enhanced Methods for Ad Hoc Dataset Retrieval

**Weiqing Luo, Qiaosheng Chen, Zhiyang Zhang, Zixian Huang, Gong Cheng**[*]

State Key Laboratory for Novel Software Technology, Nanjing University, China

{wqluo,qschen,zhiyangzhang,zixianhuang}@smail.nju.edu.cn, gcheng@nju.edu.cn

## Abstract

Ad hoc dataset retrieval has become an important way of finding data on the Web, where the underlying problem is how to measure the relevance of a dataset to a query. State-of-the-art solutions for this task are still lexical methods, which cannot capture semantic similarity. Semantics-aware knowledge-enhanced retrieval methods, which achieved promising results on other tasks, have yet to be systematically studied on this specialized task. To fill the gap, in this paper, we present an empirical investigation of the task where we implement and evaluate, on two test collections, a set of implicit and explicit knowledge-enhancement retrieval methods in various settings. Our results reveal the unique features of the task and suggest an interpolation of different kinds of methods as the current best practice.

## 1 Introduction

Tens of millions of datasets have been published on the Web (Benjelloun et al., 2020), providing government data, scientific data, etc. Accordingly, *ad hoc dataset retrieval* is becoming an important specialized information retrieval task (Kato et al., 2021; Lin et al., 2022), aiming at finding datasets that are relevant to a user's query, which is ad hoc since the number of possible queries is huge. Due to the magnitude and heterogeneity of dataset contents, existing solutions such as Google Dataset Search (Brickley et al., 2019) rely on the retrieval of dataset metadata provided by data publishers for describing their datasets, as illustrated in Table 1 which is a real example taken from the NTCIR-E test collection (Kato et al., 2021).

**Motivation.** The metadata of a dataset resembles a structured multi-field document, and current implementations of ad hoc dataset retrieval are mainly adapted from conventional document retrieval methods like BM25 or FSDM (Chapman

---

[*]Corresponding author

Table 1: A query and the metadata of a relevant dataset.

| | |
|---|---|
| **Query** | american protein intake daily |
| **Title** | USDA National Nutrient Database for Standard Reference Dataset for What We Eat In America, NHANES (Survey-SR) |
| **Description** | The dataset, Survey-SR, provides the nutrient data for assessing dietary intakes from the national survey What We Eat In America... |
| **Tags** | food-composition, food-consumption... |
| **...** | ... |

et al., 2020; Lin et al., 2022). It is well-known that such lexical retrieval methods cannot identify semantic matches and hence fail to find datasets that are lexically disjointed with but semantically relevant to the query. The problem can be alleviated by incorporating knowledge into retrieval—either *implicit* knowledge encoded in a pre-trained language model (PLM) or *explicit* knowledge stored in an encyclopedic knowledge base. For example, to identify the semantic connection between "protein" in the query and "nutrient" in the dataset in Table 1, implicit knowledge-enhanced retrieval methods may achieve it by measuring the similarity between their word embeddings, while explicit methods may employ the similarity between their linked entities in a knowledge base. However, to the best of our knowledge, *the effectiveness of incorporating knowledge into the emerging task of ad hoc dataset retrieval has not been systematically investigated*. In particular, the following three research questions remain open.

**RQ1.** Despite a few preliminary PLM-based implementations (Kato et al., 2021), they were fine-tuned on a small training set. Researches on ad hoc dataset retrieval are still in their infancy, and existing test collections provide less than 300 queries for training (Kato et al., 2021; Lin et al., 2022). The performance of PLM-based methods fine-tuned in such an in-domain setting may not be generalizable to practical settings. *Will implicit knowledge-*

*enhanced retrieval methods remain effective in ad hoc dataset retrieval in an out-of-domain setting?*

**RQ2.** The high-quality encyclopedic knowledge bases available today, like Wikipedia and Wikidata, can be used to annotate query and dataset metadata so that their semantic similarity can be measured to enhance lexical retrieval. More importantly, this is unsupervised and not limited by the availability of training data. However, such methods are so far under-studied and their effectiveness remains unknown. *Will explicit knowledge-enhanced retrieval methods be effective in ad hoc dataset retrieval?*

**RQ3.** Lexical matching, implicit knowledge, and explicit knowledge—they have the potential to capture different signals in retrieval. While each of them alone may not exhibit superb performance, their appropriate combination may produce better results, e.g., by an interpolation of their retrieval scores. *Will interpolated methods be more effective in ad hoc dataset retrieval?*

**Our Work and Contribution.** To answer the above questions, we conducted a systematic investigation of implicit and explicit knowledge-enhanced methods for ad hoc dataset retrieval on two test collections (Kato et al., 2021; Lin et al., 2022). For implicit knowledge, we evaluated five PLM-based methods in both in-domain and out-of-domain settings. For explicit knowledge, we designed and evaluated methods based on entity similarity computed over two knowledge bases. We also explored different interpolation strategies.

As the first empirical investigation of this kind, our work fills the gap and our results will provide practical guidelines for researchers and developers working with ad hoc dataset retrieval, or even information retrieval in general. It helps establish an empirical basis that will facilitate future studies on this trending information retrieval task.

Code: https://github.com/nju-websoft/AHDR-KnowledgeEnhanced

**Paper Structure.** We will discuss related work in Section 2, describe the evaluated methods in Section 3, present our experimental setup and results in Section 4 and Section 5, respectively, and finally conclude the paper in Section 6.

## 2 Related Work

### 2.1 Ad Hoc Dataset Retrieval

Ad hoc dataset retrieval is a specialized information retrieval task that aims to find the most relevant datasets to a user's query. The metadata of a dataset provided by its publisher typically consists of a set of fields such as title and description. While existing retrieval methods commonly rely on metadata (Chapman et al., 2020) which resembles a structured document, the task of dataset retrieval differs from document retrieval and has its unique properties, e.g., queries often mention geospatial and temporal entities (Kacprzak et al., 2019; Chen et al., 2019), and metadata is relatively short and often incomplete (Neumaier et al., 2016).

Knowledge-enhanced retrieval methods have the potential to exploit these features, but they have not been sufficiently studied for this task. Indeed, in a recent benchmarking effort (Lin et al., 2022), only a number of lexical retrieval methods were implemented and evaluated. In Kato et al. (2021) and Chen et al. (2023), a few PLM-based implementations were evaluated but were fine-tuned on a small training set risking overfitting and their reported performance might not be generalizable.

Our empirical investigation significantly extends the above evaluation efforts. We systematically evaluate a range of state-of-the-art PLM-based methods for ad hoc dataset retrieval in both in-domain and out-of-domain settings. Moreover, we design and evaluate explicit knowledge-enhanced methods, which are under-studied in the literature.

### 2.2 Implicit Knowledge-Enhanced Retrieval

PLMs encode knowledge into learnable dense vectors (Talmor et al., 2020). PLM-based retrieval, aka dense retrieval, has developed rapidly in recent years and exhibited powerful text understanding capabilities which helped improve the accuracy of document retrieval (Zhao et al., 2022). Among others, monoBERT (Nogueira and Cho, 2019) directly leverages the text classification capability of BERT (Devlin et al., 2019) to rank documents. DPR (Karpukhin et al., 2020) adopts a dual-encoder architecture that employs the implicit knowledge in PLM and performs metric learning. Other dense retrieval models such as ColBERT (Khattab and Zaharia, 2020) and COIL (Gao et al., 2021) further exploit the implicit knowledge in PLMs by computing token-level matching through multiple vectors. Xiong et al. (2021) proposes ANCE which features dynamic negative sampling to improve the informativeness of training data. Condenser (Gao and Callan, 2021) adopts a novel pre-training architecture to compress information in the text. Further, coCondenser (Gao and

Callan, 2022) extends Condenser by pre-training with a query-agnostic contrastive loss.

These state-of-the-art dense retrieval methods have not been applied to ad hoc dataset retrieval. We adapt them to this new task and thoroughly analyze their effectiveness in various settings.

### 2.3 Explicit Knowledge-Enhanced Retrieval

Since the introduction of the Semantic Web, it has benefited information retrieval systems by incorporating explicit knowledge. In McCool and Miller (2003), an early semantic search prototype named TAP was presented, showing that knowledge bases can enhance retrieval systems. For Web search, Lu et al. (2009) demonstrated that ranking methods can be improved by semantic features. For entity search, researchers have employed entity linking techniques to annotate queries and measured the semantic relevance of a target entity to a query based on their learned embeddings (Gerritse et al., 2020).

Such explicit knowledge-enhanced retrieval methods have not received much attention in the research of ad hoc dataset retrieval. We design a method for this new task and evaluate its various configurations using different knowledge bases, entity linking tools, and entity embeddings.

## 3 Methods

We divide knowledge-enhanced methods for ad hoc dataset retrieval into two types: using implicit knowledge and using explicit knowledge. Employing implicit knowledge embodied in PLMs to enhance retrieval has been widely used, which we will briefly review in Section 3.1. In Section 3.2 we will present a method for employing explicit knowledge in a knowledge base to enhance retrieval.

**Problem Statement.** Given a query $q$, the main task in ad hoc dataset retrieval is to compute the relevance of each dataset $d$ to $q$ denoted by $\mathrm{rel}(d, q)$, so that a ranked list of datasets can be returned.

### 3.1 Implicit Knowledge-Enhanced Retrieval

While the contents of different datasets can be in different formats (e.g., TXT, CSV, JSON), the metadata of a dataset typically consists of a set of fields such as title and description which are commonly used in retrieval. For each dataset $d$, we concatenate the textual values of its metadata fields $\{T_{d,1}, T_{d,2}, \ldots\}$ into a document $T_d$:

$$T_d = T_{d,1} \oplus T_{d,2} \oplus \cdots, \qquad (1)$$

where $\oplus$ represents concatenation.

Any dense retriever reviewed in Section 2.2 can be used to compute the relevance of $T_d$ to the query $q$ as the relevance score of $d$:

$$\mathrm{rel}(d, q) = \mathrm{DenseRetrieval}(T_d, q), \qquad (2)$$

which exploits knowledge implicitly encoded by PLMs into learnable dense vectors. Dense retrievers are commonly supervised. Fine-tuning can be performed on task-specific training data—referred to as an in-domain setting, or on data for other tasks—referred to as an out-of-domain setting.

### 3.2 Explicit Knowledge-Enhanced Retrieval

We assume that explicit knowledge is given as a knowledge base describing a set of entities $E$. To exploit such knowledge, we firstly link the query $q$ and the document representation $T_d$ for each dataset $d$ to two sets of entities in $E$, denoted by $E_q \subseteq E$ and $E_d \subseteq E$, respectively. Then we aggregate the pairwise entity similarities between $E_q$ and $E_d$ as the relevance of $d$ to $q$. Entity similarity is measured based on their textual and structural descriptions in the knowledge base.

#### 3.2.1 Entity Linking

We link $q$ to a set of entities $E_q \subseteq E$ that are mentioned in $q$ to represent $q$. Analogously, we link $T_d$ to a set of entities $E_d \subseteq E$ that are mentioned in $T_d$ to represent $d$. Entity linking is an established research problem (Shen et al., 2015, 2023) and we use off-the-shelf tools in the experiments.

#### 3.2.2 Entity Set Similarity

Let $\mathrm{esim}(e_i, e_j)$ be the similarity between two entities $e_i, e_j \in E$, which will be elaborated in Section 3.2.3. Let $\mathbf{S}$ be an $m \times n$ dimensional similarity matrix containing $m = |E_q|$ rows and $n = |E_d|$ columns. For $1 \leq i \leq m$ and $1 \leq j \leq n$, each element $s_{i,j}$ represents the entity similarity between $e_i \in E_q$ and $e_j \in E_d$, i.e., $s_{i,j} = \mathrm{esim}(e_i, e_j)$. We aggregate the similarity values in $\mathbf{S}$ as follows, also depicted in Figure 1.

For each entity $e_i \in E_q$, we find its most similar entity in $E_d$ and take their similarity value. We calculate the arithmetic mean (arithmean) of such similarity values over all the $m$ entities in $E_q$:

$$s^{\mathrm{row}} = \mathrm{arithmean}\{\max_{1 \leq j \leq n} s_{i,j} \mid 1 \leq i \leq m\}, \quad (3)$$

Analogously, we compute to what extent the entities in $E_q$ can best "cover" the entities in $E_d$:

$$s^{\mathrm{col}} = \mathrm{arithmean}\{\max_{1 \leq i \leq m} s_{i,j} \mid 1 \leq j \leq n\}. \quad (4)$$

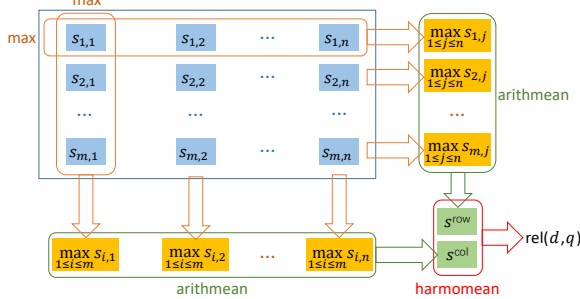

Figure 1: Aggregation of similarity matrix.

Finally, considering that two similar sets of entities should both largely cover each other, we calculate the harmonic mean (harmomean) of $s^{\text{row}}$ and $s^{\text{col}}$, which aggregates the similarity values in $\mathbf{S}$ as the relevance score of $d$:

$$\text{rel}(d, q) = \text{harmomean}(s^{\text{row}}, s^{\text{col}}) . \quad (5)$$

By the property of harmonic mean, $\text{rel}(d, q)$ will be high only if both $s^{\text{row}}$ and $s^{\text{col}}$ are high.

### 3.2.3 Entity Similarity

Now we elaborate $\text{esim}(e_i, e_j)$, the similarity between two entities $e_i, e_j \in E$. We measure and integrate their entity- and word-level similarities.

First, we measure the entity-level similarity between $e_i$ and $e_j$ based on their embedding vectors. Representation learning is an established research problem (Wang et al., 2017; Yang et al., 2022), which encodes the textual and structural description of each entity in a knowledge base into a dense vector. Let $\mathbf{e}_i, \mathbf{e}_j$ be the embeddings of $e_i, e_j$, respectively. We calculate their cosine similarity:

$$s_{i,j}^{\text{ent}} = \cos(\mathbf{e}_i, \mathbf{e}_j) . \quad (6)$$

Second, we measure the word-level similarity between $e_i$ and $e_j$ based on the embedding vectors of their mentions. Specifically, let $W_i, W_j$ be the sets of words that appear in the mentions of $e_i, e_j$, respectively. We construct a $|W_i| \times |W_j|$ dimensional similarity matrix where each element represents the cosine similarity between two word embedding vectors. Then we aggregate the similarity values in the matrix in a way that resembles the aggregation process described in Section 3.2.2 and Figure 1. The result is denoted by $s_{i,j}^{\text{word}}$.

Finally, we integrate entity- and word-level similarities by calculating their harmonic mean:

$$\text{esim}(e_i, e_j) = \text{harmomean}(s_{i,j}^{\text{ent}}, s_{i,j}^{\text{word}}) . \quad (7)$$

We choose harmonic mean because it empirically outperforms several other combination functions such as arithmetic mean, maximum, and minimum. Table 7 in the appendix presents the results of using different combination functions.

## 4 Experimental Setup

### 4.1 Test Collections

We conducted experiments on two test collections for ad hoc dataset retrieval.

#### 4.1.1 NTCIR-E

NTCIR-E[1] is the English version of the test collection used in the NTCIR-15 Dataset Search task (Kato et al., 2021), including 46,615 datasets and 192 queries. The datasets were collected from Data.gov. The queries were crowdsourced, and were originally split into 96 as the training set and 96 as the test set. We further split the former into 76 as our training set and 20 as our validation set. The relevance of a dataset to a query has been annotated as irrelevant (0), partially-relevant (1), or relevant (2) as the gold standard.

#### 4.1.2 ACORDAR

ACORDAR[2] is a test collection specifically over RDF datasets (Lin et al., 2022), including 31,589 datasets and 493 queries. The datasets were collected from 543 open data portals. The queries were partially crowdsourced and partially collected from TREC, and were split into five subsets for five-fold cross-validation, each fold using three subsets as the training set, one subset as the validation set, and one subset as the test set. Similar to NTCIR-E, the relevance of a dataset to a query has been annotated as irrelevant (0), partially-relevant (1), or relevant (2) as the gold standard.

### 4.2 Evaluation Metrics

Retrieval on NTCIR-E relied on two metadata fields: title and description. ACORDAR further included two other fields: author and tags. Both NTCIR-E and ACORDAR have provided top-10 retrieval results returned by (field-weighted) BM25 over these fields. Based on these first-stage retrieval results, we investigated the **reranking** performance of knowledge-enhanced retrieval methods.

Two evaluation metrics were used: normalized discounted cumulative gain (**NDCG**) and mean

---

[1]https://ntcir.datasearch.jp/data_search_1/
[2]https://github.com/nju-websoft/ACORDAR

average precision (**MAP**). When calculating MAP scores, both partially-relevant and relevant in the gold standard were considered as relevant.

### 4.3 Implementation Details

#### 4.3.1 Implicit Knowledge-Enhanced Retrieval

We used five popular PLM-based methods for our experiments: **monoBERT**-large (Nogueira and Cho, 2019),[3] **monoT5**-large (Nogueira et al., 2020),[4] **coCondenser** (Gao and Callan, 2022),[5] **ColBERT**-v2 (Khattab and Zaharia, 2020),[6] and **ANCE** (Xiong et al., 2021)[7]. For each model, we used its checkpoint pre-trained on MS MARCO (Nguyen et al., 2016), and reported its performance on the test set of each test collection to measure its performance in an out-of-domain (**OOD**) setting. We also fine-tuned it on the training and validation sets of each test collection and then measured its performance on the test set in an in-domain (**ID**) setting, where partially-relevant and relevant datasets in the training set were used as positive samples, and irrelevant datasets were used as negative samples. It is worth noting that we fine-tuned and used all these methods as black boxes, e.g., ANCE's special negative sampling strategy as well as all the other special strategies incorporated into these methods were executed.

For hyperparameter tuning in the in-domain setting, monoBERT and monoT5 were tuned with learning rate from {3e-5, 5e-5, 1e-4} and batch size from {16, 24}. coCondenser, ColBERT, and ANCE were tuned with learning rate from {1e-6, 3e-6, 5e-6} and batch size from {32, 64}, {8, 16}, and {4, 8}, respectively. Each model was trained for 10 epochs. The best-performing configuration on the validation set was selected to run on the test set. We used NVIDIA GeForce RTX 3090 GPUs.

#### 4.3.2 Explicit Knowledge-Enhanced Retrieval

We used two well-known encyclopedic knowledge bases, **Wikipedia** and **Wikidata**, and used their corresponding entity linking tools and embeddings.

For Wikipedia, we used **TAGME** (Ferragina and Scaiella, 2010)[8] for entity linking; we also reported

the results using **REL** (Van Hulst et al., 2020)[9] for comparison. We collected entity embeddings from **Wikipedia2vec** (Yamada et al., 2018).[10]

For Wikidata, we used **Falcon 2.0** (Sakor et al., 2020)[11] for entity linking. We collected entity embeddings from **KGTK** (Ilievski et al., 2021)[12] (the `text` version); we also reported the results using **RDF2vec** (Ristoski et al., 2019)[13] (the `sg_200_5_5_15_4_500` version) for comparison.

For word embeddings, we consistently collected from Wikipedia2vec.

## 5 Experimental Results

Each of the following three subsections reported experimental results to answer one of the three research questions raised in Section 1.

### 5.1 Implicit Knowledge-Enhanced Retrieval

As shown in Table 2, in an in-domain setting, reranking by coCondenser, ColBERT, and ANCE achieved significant improvements on both test collections in terms of NDCG@5. However, for monoBERT the improvement was marginal on NTCIR-E, and for monoT5 we even observed a performance drop on ACORDAR. The results indicated that *the fine-tuned PLM-based methods might have overfitted the small training sets of existing test collections for ad hoc dataset retrieval*. monoBERT and monoT5 performed extensive query-document interaction, probably leading to more severe overfitting and worse test results.

The out-of-domain setting analyzed the generalizability of PLM-based methods. While reranking by monoT5, coCondenser, and ANCE achieved significant improvements on NTCIR-E in terms of NDCG@5, only monoT5 also achieved a significant improvement on ACORDAR, and for ColBERT we even observed a performance drop on both test collections. The results suggested that *only a few domain-adapted PLM-based methods could directly generalize to the task of ad hoc dataset retrieval having its unique features*. ColBERT which relies on word-level matching achieved relatively poor results possibly due to some ambiguous query words such as 'MCI' which could be mistakenly matched with the same word

---

[3]https://huggingface.co/castorini/monobert-large-msmarco

[4]https://huggingface.co/castorini/monot5-large-msmarco

[5]https://huggingface.co/Luyu/co-condenser-marco

[6]https://downloads.cs.stanford.edu/nlp/data/colbert/colbertv2/colbertv2.0.tar.gz

[7]https://huggingface.co/castorini/ance-msmarco-passage

[8]https://tagme.d4science.org/tagme/

[9]https://github.com/informagi/REL

[10]https://wikipedia2vec.github.io/wikipedia2vec/

[11]https://github.com/SDM-TIB/falcon2.0

[12]https://github.com/usc-isi-i2/kgtk-similarity

[13]https://data.dws.informatik.uni-mannheim.de/rdf2vec/models/Wikidata/4depth/skipgram/

Table 2: Performance of implicit knowledge-enhancement retrieval methods, with $^*$ indicating a significant improvement after reranking (according to paired t-test under $p < 0.05$).

| Test collection | Reranking method | NDCG@5 | NDCG@10 | MAP@5 | MAP@10 |
|---|---|---|---|---|---|
| NTCIR-E | before reranking | 0.2252 | 0.2385 | 0.1232 | 0.1556 |
| | monoBERT (ID) | 0.2280 | 0.2364 | 0.1188 | 0.1494 |
| | monoBERT (OOD) | 0.2554 | 0.2513 | 0.1361 | 0.1645 |
| | monoT5 (ID) | 0.2532 | 0.2497 | 0.1415 | 0.1694 |
| | monoT5 (OOD) | 0.2833$^*$ | 0.2702$^*$ | 0.1593$^*$ | 0.1852$^*$ |
| | coCondenser (ID) | 0.2903$^*$ | 0.2760$^*$ | 0.1756$^*$ | 0.1981$^*$ |
| | coCondenser (OOD) | 0.2933$^*$ | 0.2775$^*$ | 0.1785$^*$ | 0.2001$^*$ |
| | ColBERT (ID) | 0.2664$^*$ | 0.2604 | 0.1522 | 0.1797 |
| | ColBERT (OOD) | 0.2121 | 0.2217 | 0.1047 | 0.1351 |
| | ANCE (ID) | 0.3024$^*$ | 0.2794$^*$ | 0.1803$^*$ | 0.2023$^*$ |
| | ANCE (OOD) | 0.2764$^*$ | 0.2650 | 0.1601 | 0.1851 |
| ACORDAR | before reranking | 0.5045 | 0.5249 | 0.2859 | 0.3837 |
| | monoBERT (ID) | 0.5521$^*$ | 0.5511$^*$ | 0.3239$^*$ | 0.4130$^*$ |
| | monoBERT (OOD) | 0.5133 | 0.5305 | 0.2855 | 0.3835 |
| | monoT5 (ID) | 0.3570 | 0.4418 | 0.1904 | 0.3056 |
| | monoT5 (OOD) | 0.5262$^*$ | 0.5338 | 0.3049$^*$ | 0.3981 |
| | coCondenser (ID) | 0.5300$^*$ | 0.5360 | 0.3076$^*$ | 0.3998$^*$ |
| | coCondenser (OOD) | 0.5144 | 0.5259 | 0.2963 | 0.3901 |
| | ColBERT (ID) | 0.5273$^*$ | 0.5372 | 0.3080$^*$ | 0.4010$^*$ |
| | ColBERT (OOD) | 0.4451 | 0.4889 | 0.2430 | 0.3473 |
| | ANCE (ID) | 0.5327$^*$ | 0.5416$^*$ | 0.3123$^*$ | 0.4048$^*$ |
| | ANCE (OOD) | 0.5125 | 0.5289 | 0.2892 | 0.3850 |

Table 3: Performance of explicit knowledge-enhancement retrieval methods, with $^*$ indicating a significant improvement after reranking (according to paired t-test under $p < 0.05$).

| Test collection | Reranking method | NDCG@5 | NDCG@10 | MAP@5 | MAP@10 |
|---|---|---|---|---|---|
| NTCIR-E | before reranking | 0.2252 | 0.2385 | 0.1232 | 0.1556 |
| | Wikipedia w/ TAGME | 0.2661$^*$ | 0.2576 | 0.1521 | 0.1782 |
| | Wikipedia w/ REL | 0.2241 | 0.2378 | 0.1227 | 0.1551 |
| | Wikidata w/ KGTK | 0.2314 | 0.2392 | 0.1211 | 0.1521 |
| | Wikidata w/ RDF2vec | 0.2135 | 0.2337 | 0.1191 | 0.1518 |
| ACORDAR | before reranking | 0.5045 | 0.5249 | 0.2859 | 0.3837 |
| | Wikipedia w/ TAGME | 0.4963 | 0.5150 | 0.2746 | 0.3725 |
| | Wikipedia w/ REL | 0.4636 | 0.5002 | 0.2519 | 0.3563 |
| | Wikidata w/ KGTK | 0.4682 | 0.4968 | 0.2552 | 0.3558 |
| | Wikidata w/ RDF2vec | 0.4607 | 0.4964 | 0.2459 | 0.3516 |

having a different meaning in an irrelevant document. Other models relying on the global [CLS] token representation were less influenced by such individual word mismatches.

## 5.2 Explicit Knowledge-Enhanced Retrieval

As shown in Table 3, only reranking by using Wikipedia knowledge with TAGME achieved a significant improvement on NTCIR-E in terms of NDCG@5. We observed performance drops in all the other configurations, and observed noticeable differences between the performance of using different entity linking tools (i.e., TAGME or REL) and different entity embeddings (i.e., KGTK or RDF2vec). The results suggested that *reranking with explicit knowledge could have the potential but also require very careful implementation to obtain effectiveness in ad hoc dataset retrieval.*

More concretely, we attributed the relatively good performance of TAGME to its higher recall than REL. Indeed, the sparse Wikipedia links found by REL could not sufficiently capture the semantics of the original text. RDF2vec which mainly employed the graph structure of Wikidata was inferior to KGTK whose text version used in our experiments ignored graph structure and only exploited the textual description of entities, which seemed to be more helpful than the graph structure.

## 5.3 Score Interpolation

### 5.3.1 Interpolation with BM25 Scores

In Section 5.1 and Section 5.2, we directly used the relevance scores computed by knowledge-enhanced retrieval methods to rerank first-stage retrieval results. Their unsatisfying results might be partially related to their weak sensitivity to exact lexical matches. Therefore, a straightforward extension would be to interpolate their score with BM25 score to enhance their capability of lexical matching, i.e., by calculating the sum of the two scores (both after min-max normalization). We chose sum because it was empirically among the best-performing fusion algorithms for interpolation. Table 8 in the appendix presents the results of using different fusion algorithms provided by ranx (Bassani and Romelli, 2022).[14]

For implicit knowledge-enhanced retrieval methods, as shown in Table 4, interpolation helped improve the performance of all the methods on ACORDAR in an out-of-domain setting, and all

---

[14]https://amenra.github.io/ranx/fusion/

the improvements were significant in terms of NDCG@5 (represented by †). On NTCIR-E, interpolation improved the performance of monoBERT and ColBERT, although it worsened the performance of the other methods. With interpolation, reranking by almost all the methods (except for ColBERT) achieved significant improvements on both test collections in terms of NDCG@5 (represented by *), whereas without interpolation only monoT5 achieved that. The results demonstrated that *domain-adapted PLM-based methods interpolated with BM25 scores could effectively generalize to ad hoc dataset retrieval.*

For explicit knowledge-enhanced retrieval methods, as shown in Table 5, interpolation helped improve the performance of all the methods on ACORDAR, and all the improvements were significant (represented by †). On NTCIR-E, interpolation also noticeably improved the performance of using Wikidata knowledge, while its influence on the methods using Wikipedia knowledge was minor. With interpolation, reranking by using Wikipedia knowledge with TAGME achieved significant improvements on both test collections in terms of NDCG@5 (represented by *), whereas without interpolation it achieved that only on NTCIR-E. The results demonstrated that *reranking with explicit knowledge interpolated with BM25 scores could effectively benefit ad hoc dataset retrieval.*

### 5.3.2 Interpolation of Implicit and Explicit Knowledge-Enhanced Retrieval

In Section 5.3.1, reranking with implicit and explicit knowledge interpolated with BM25 scores both showed effectiveness, so we continued to explore whether the two kinds of knowledge could complement each other by further interpolating implicit knowledge-enhanced relevance score, explicit knowledge-enhanced relevance score, and BM25 score, i.e., by calculating the sum of the three scores (all after min-max normalization). For a focused discussion, we reported the results of monoT5 and Wikipedia with TAGME, the best-performing implicit and explicit knowledge-enhanced retrieval methods interpolated with BM25 scores according to Table 4 and Table 5, respectively. Similar results were observed on the other methods and hence omitted.

As shown in Table 6, such further interpolation achieved the highest scores in all the settings. In particular, compared with using explicit knowledge interpolated with BM25 scores, incorporating im-

Table 4: Performance of implicit knowledge-enhanced retrieval methods interpolated with BM25 scores, with * indicating a significant improvement after reranking, and † indicating a significant improvement after interpolation with BM25 scores (paired t-test with $p < 0.05$).

| Test collection | Reranking method | NDCG@5 | NDCG@10 | MAP@5 | MAP@10 |
|---|---|---|---|---|---|
| NTCIR-E | before reranking | 0.2252 | 0.2385 | 0.1232 | 0.1556 |
| | monoBERT (OOD) | 0.2554 | 0.2513 | 0.1361 | 0.1645 |
| | interpolated with BM25 | $0.2582^*$ | 0.2569 | 0.1422 | 0.1711 |
| | monoT5 (OOD) | $0.2833^*$ | $0.2702^*$ | $0.1593^*$ | $0.1852^*$ |
| | interpolated with BM25 | $0.2697^*$ | $0.2616^*$ | $0.1498^*$ | $0.1778^*$ |
| | coCondenser (OOD) | $0.2933^*$ | $0.2775^*$ | $0.1785^*$ | $0.2001^*$ |
| | interpolated with BM25 | $0.2693^*$ | $0.2638^*$ | $0.1600^*$ | $0.1855^*$ |
| | ColBERT (OOD) | 0.2121 | 0.2217 | 0.1047 | 0.1351 |
| | interpolated with BM25 | 0.2361 | 0.2467† | 0.1334 | 0.1657† |
| | ANCE (OOD) | $0.2764^*$ | 0.2650 | 0.1601 | 0.1851 |
| | interpolated with BM25 | $0.2648^*$ | $0.2555^*$ | $0.1486^*$ | $0.1748^*$ |
| ACORDAR | before reranking | 0.5045 | 0.5249 | 0.2859 | 0.3837 |
| | monoBERT (OOD) | 0.5133 | 0.5305 | 0.2855 | 0.3835 |
| | interpolated with BM25 | $0.5324^*$† | $0.5398^*$† | $0.3011^*$† | $0.3952^*$† |
| | monoT5 (OOD) | $0.5262^*$ | 0.5338 | $0.3049^*$ | 0.3981 |
| | interpolated with BM25 | $0.5496^*$† | $0.5499^*$† | $0.3212^*$† | $0.4126^*$† |
| | coCondenser (OOD) | 0.5144 | 0.5259 | 0.2963 | 0.3901 |
| | interpolated with BM25 | $0.5378^*$† | $0.5386^*$† | $0.3101^*$† | $0.4000^*$ |
| | ColBERT (OOD) | 0.4451 | 0.4889 | 0.2430 | 0.3473 |
| | interpolated with BM25 | 0.5003† | 0.5210† | 0.2833† | 0.3800† |
| | ANCE (OOD) | 0.5125 | 0.5289 | 0.2892 | 0.3850 |
| | interpolated with BM25 | $0.5368^*$† | $0.5427^*$† | $0.3149^*$† | $0.4056^*$† |

Table 5: Performance of explicit knowledge-enhanced retrieval methods interpolated with BM25 scores, with * indicating a significant improvement after reranking, and † indicating a significant improvement after interpolation with BM25 scores (paired t-test with $p < 0.05$).

| Test collection | Reranking method | NDCG@5 | NDCG@10 | MAP@5 | MAP@10 |
|---|---|---|---|---|---|
| NTCIR-E | before reranking | 0.2252 | 0.2385 | 0.1232 | 0.1556 |
| | Wikipedia w/ TAGME | $0.2661^*$ | 0.2576 | 0.1521 | 0.1782 |
| | interpolated with BM25 | $0.2634^*$ | $0.2565^*$ | $0.1490^*$ | $0.1755^*$ |
| | Wikipedia w/ REL | 0.2241 | 0.2378 | 0.1227 | 0.1551 |
| | interpolated with BM25 | 0.2252 | 0.2385 | 0.1232 | 0.1556 |
| | Wikidata w/ KGTK | 0.2314 | 0.2392 | 0.1211 | 0.1521 |
| | interpolated with BM25 | 0.2590 | 0.2557 | 0.1468 | 0.1737 |
| | Wikidata w/ RDF2vec | 0.2135 | 0.2337 | 0.1191 | 0.1518 |
| | interpolated with BM25 | 0.2310 | 0.2435 | 0.1328 | 0.1634 |
| ACORDAR | before reranking | 0.5045 | 0.5249 | 0.2859 | 0.3837 |
| | Wikipedia w/ TAGME | 0.4963 | 0.5150 | 0.2746 | 0.3725 |
| | interpolated with BM25 | $0.5243^*$† | $0.5347^*$† | $0.3003^*$† | $0.3952^*$† |
| | Wikipedia w/ REL | 0.4636 | 0.5002 | 0.2519 | 0.3563 |
| | interpolated with BM25 | 0.4987† | 0.5218† | 0.2801† | 0.3794† |
| | Wikidata w/ KGTK | 0.4682 | 0.4968 | 0.2552 | 0.3558 |
| | interpolated with BM25 | $0.5103^*$† | $0.5274^*$† | $0.2912^*$† | $0.3885^*$† |
| | Wikidata w/ RDF2vec | 0.4607 | 0.4964 | 0.2459 | 0.3516 |
| | interpolated with BM25 | 0.5092† | 0.5277† | 0.2885† | 0.3873† |

Table 6: Performance of interpolation of implicit and explicit knowledge-enhanced retrieval methods, with * indicating a significant improvement after reranking, † and ‡ indicating a significant improvement after incorporating explicit and implicit knowledge, respectively (paired t-test with $p < 0.05$).

| Test collection | Reranking method | NDCG@5 | NDCG@10 | MAP@5 | MAP@10 |
|---|---|---|---|---|---|
| NTCIR-E | before reranking | 0.2252 | 0.2385 | 0.1232 | 0.1556 |
| | monoT5 (OOD), interpolated with BM25 | $0.2697^*$ | $0.2616^*$ | $0.1498^*$ | $0.1778^*$ |
| | Wikipedia w/ TAGME, interpolated with BM25 | $0.2634^*$ | $0.2565^*$ | $0.1490^*$ | $0.1755^*$ |
| | interpolation of monoT5 (OOD), Wikipedia w/ TAGME, and BM25 | $0.2827^*$ | $0.2676^*$ | $0.1631^*$ | $0.1870^*$ |
| ACORDAR | before reranking | 0.5045 | 0.5249 | 0.2859 | 0.3837 |
| | monoT5 (OOD), interpolated with BM25 | $0.5496^*$ | $0.5499^*$ | $0.3212^*$ | $0.4126^*$ |
| | Wikipedia w/ TAGME, interpolated with BM25 | $0.5243^*$ | $0.5347^*$ | $0.3003^*$ | $0.3952^*$ |
| | interpolation of monoT5 (OOD), Wikipedia w/ TAGME, and BM25 | $0.5570^*$‡ | $0.5523^*$‡ | $0.3244^*$‡ | $0.4139^*$‡ |

plicit knowledge brought significant improvements on ACORDAR (represented by ‡), while their complementarity in the other settings was not strong. The results suggested that *using both implicit and explicit knowledge interpolated with first-stage lexical retrieval scores could represent the current best practice in reranking for ad hoc dataset retrieval*.

## 6   Conclusion and Future Work

We summarize our empirical findings to answer the three research questions raised in Section 1.

*RQ1: Will implicit knowledge-enhanced retrieval methods remain effective in ad hoc dataset retrieval in an out-of-domain setting?* According to the results presented in Section 5.1, only monoT5 remained effective on its own, whereas other PLM-based methods could not consistently bring improvements. However, interpolated with BM25 scores, most of these methods effectively generalized to this new task as shown by the results in Section 5.3.1. It demonstrates the necessity of combining dense and sparse retrieval for this task.

*RQ2: Will explicit knowledge-enhanced retrieval methods be effective in ad hoc dataset retrieval?* According to the results presented in Section 5.2 and Section 5.3.1, reranking with explicit knowledge could be beneficial to this task only when performing interpolation with BM25 scores, and significant improvements were consistently observed only for the configuration using Wikipedia knowledge with TAGME for entity linking. It suggests that the incorporation of explicit knowledge into this task should be carefully designed, and calls for more effective implementations in the future.

*RQ3: Will interpolated methods be more effective in ad hoc dataset retrieval?* According to the results presented in Section 5.3.1 and Section 5.3.2, not only interpolation with BM25 scores was helpful, but also implicit and explicit knowledge exhibited complementarity. A combination of lexical matching, implicit knowledge, and explicit knowledge consistently achieved the best performance in our experiments, representing the current best practice for solving this task.

Hopefully our empirical findings and conclusions should provide useful guidelines for the community to research and practice ad hoc dataset retrieval. In future work, we will continue to explore and address the unique challenges of this important task. Regarding implicit knowledge-enhanced retrieval, we are interested in constructing a large

test collection to support in-domain supervision of more generalizable PLM-based retrieval methods. We also plan to apply PLM-based methods to not only dataset metadata but also dataset contents which are large and heterogeneous, thus posing great challenges. A related trending research direction is to explore the effectiveness of large language models in ad hoc dataset retrieval. Regarding explicit knowledge-enhanced retrieval, since the accuracy of entity linking observed in our experiments was not satisfactory and might distort the results, an idea is to extend existing test collections with manually annotated entities, which would also provide a useful resource for entity linking research.

## Acknowledgements

This work was supported by the NSFC (62072224).

## Limitations

First, although we have already used two test collections for ad hoc dataset retrieval, there was the possibility that our results were still biased due to the small number of queries (less than a few hundred) in these test collections. The generalizability of our conclusions could be improved in the future when new and larger test collections are available.

Second, some observations in our experiments have yet to be justified. For example, while reranking by most PLM-based methods interpolated with BM25 scores showed effectiveness, it remains unclear why ColBERT was an exception. Exploring such reasons could help deepen our understanding of this task as well as the strengths and weaknesses of existing retrieval methods.

Third, our score interpolation performed in the experiments was helpful but simple. There could be more effective interpolation strategies. For example, instead of score interpolation, implicit and explicit knowledge could be integrated into a single model. We have witnessed the emergence of such efforts, which will be evaluated in our future work.

Fourth, following common practice in the literature, we only considered dataset metadata in retrieval without using dataset contents due to their magnitude and heterogeneity, which could not be effectively handled by current PLMs. It remains an open question as to whether and how dataset content should be exploited in retrieval.

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

# A  Appendix: Additional Experiments

Table 7: Performance of explicit knowledge-enhanced retrieval (Wikipedia w/ TAGME) using different functions for combining entity- and word-level similarities.

| Test collection | Combination | NDCG@5 | NDCG@10 | MAP@5 | MAP@10 |
|---|---|---|---|---|---|
| NTCIR-E | arithmean | 0.2641 | **0.2578** | 0.1511 | 0.1779 |
| | harmomean | **0.2661** | 0.2576 | **0.1521** | **0.1782** |
| | geomemean | **0.2661** | 0.2576 | **0.1521** | 0.1781 |
| | max | 0.2472 | 0.2450 | 0.1317 | 0.1619 |
| | min | 0.2538 | 0.2524 | 0.1434 | 0.1719 |
| ACORDAR | arithmean | 0.4936 | 0.5149 | **0.2752** | **0.3735** |
| | harmomean | **0.4963** | **0.5150** | 0.2746 | 0.3725 |
| | geomemean | 0.4948 | 0.5144 | 0.2739 | 0.3719 |
| | max | 0.4767 | 0.5019 | 0.2586 | 0.3585 |
| | min | 0.4900 | 0.5117 | 0.2707 | 0.3697 |

Table 8: Performance of implicit knowledge-enhanced retrieval methods interpolated with BM25 scores using different fusion algorithms for interpolation.

| Test collection | Reranking method | Fusion | NDCG@5 | NDCG@10 | MAP@5 | MAP@10 |
|---|---|---|---|---|---|---|
| NTCIR-E | monoBERT (ID) interpolated with BM25 | sum | 0.2494 | 0.2561 | 0.1408 | 0.1707 |
| | | rrf | 0.2501 | 0.2499 | 0.1389 | 0.1666 |
| | | wmnz | 0.2494 | 0.2561 | 0.1408 | 0.1707 |
| | | wsum | **0.2609** | **0.2618** | **0.1498** | **0.1781** |
| | monoT5 (ID) interpolated with BM25 | sum | **0.2621** | 0.2553 | **0.1502** | 0.1760 |
| | | rrf | 0.2457 | 0.2498 | 0.1361 | 0.1658 |
| | | wmnz | **0.2621** | 0.2553 | **0.1502** | 0.1760 |
| | | wsum | 0.2583 | **0.2568** | 0.1484 | **0.1771** |
| | coCondenser (ID) interpolated with BM25 | sum | **0.2794** | **0.2709** | **0.1705** | **0.1943** |
| | | rrf | 0.2695 | 0.2609 | 0.1583 | 0.1824 |
| | | wmnz | **0.2794** | **0.2709** | **0.1705** | **0.1943** |
| | | wsum | 0.2252 | 0.2385 | 0.1232 | 0.1556 |
| | ColBERT (ID) interpolated with BM25 | sum | **0.2741** | **0.2685** | **0.1608** | **0.1883** |
| | | rrf | 0.2657 | 0.2598 | 0.1503 | 0.1784 |
| | | wmnz | **0.2741** | **0.2685** | **0.1608** | **0.1883** |
| | | wsum | 0.2673 | 0.2628 | 0.1520 | 0.1804 |
| | ANCE (ID) interpolated with BM25 | sum | 0.2864 | 0.2693 | 0.1649 | 0.1891 |
| | | rrf | 0.2687 | 0.2656 | 0.1517 | 0.1815 |
| | | wmnz | 0.2864 | 0.2693 | 0.1649 | 0.1891 |
| | | wsum | **0.3024** | **0.2766** | **0.1803** | **0.2010** |
| ACORDAR | monoBERT (ID) interpolated with BM25 | sum | **0.5625** | **0.5563** | **0.3307** | **0.4191** |
| | | rrf | 0.5534 | 0.5504 | 0.3241 | 0.4124 |
| | | wmnz | 0.5616 | 0.5557 | 0.3303 | 0.4187 |
| | | wsum | 0.5305 | 0.5388 | 0.3066 | 0.3993 |
| | monoT5 (ID) interpolated with BM25 | sum | 0.4015 | 0.4657 | 0.2242 | 0.3323 |
| | | rrf | 0.4236 | 0.4701 | 0.2288 | 0.3338 |
| | | wmnz | 0.4003 | 0.4649 | 0.2238 | 0.3319 |
| | | wsum | **0.4966** | **0.5195** | **0.2846** | **0.3820** |
| | coCondenser (ID) interpolated with BM25 | sum | **0.5502** | **0.5465** | **0.3206** | **0.4092** |
| | | rrf | 0.5403 | 0.5423 | 0.3141 | 0.4042 |
| | | wmnz | 0.5493 | 0.5459 | 0.3202 | 0.4087 |
| | | wsum | 0.5156 | 0.5278 | 0.2984 | 0.3895 |
| | ColBERT (ID) interpolated with BM25 | sum | **0.5472** | **0.5486** | **0.3223** | **0.4132** |
| | | rrf | 0.5390 | 0.5403 | 0.3107 | 0.4014 |
| | | wmnz | 0.5463 | 0.5481 | 0.3219 | 0.4129 |
| | | wsum | 0.5087 | 0.5247 | 0.2898 | 0.3851 |
| | ANCE (ID) interpolated with BM25 | sum | **0.5583** | **0.5552** | **0.3320** | **0.4196** |
| | | rrf | 0.5419 | 0.5425 | 0.3160 | 0.4050 |
| | | wmnz | 0.5574 | 0.5546 | 0.3317 | 0.4192 |
| | | wsum | 0.5076 | 0.5248 | 0.2902 | 0.3859 |