# OpenReview forum: "An Empirical Investigation of Implicit and Explicit Knowledge-Enhanced Methods for Ad Hoc Dataset Retrieval"
_EMNLP/2023/Conference — EMNLP 2023 Findings_

### Official Review · Reviewer_Afe6 · 2023-08-04

**Soundness:** 4

**Excitement:**

3: Ambivalent: It has merits (e.g., it reports state-of-the-art results, the idea is nice), but there are key weaknesses (e.g., it describes incremental work), and it can significantly benefit from another round of revision. However, I won't object to accepting it if my co-reviewers champion it.

**Missing References:**

I argue that the paper should feature more in-depth discussions about (a) other methods for representing text with dense embeddings, and (b) methods for combining results of different retrieval methods. The experimental results should ideally reflect these other studies as well.

A - The experiments with neural/dense retrieval models focused on approaches pre-trained on MS-MARCO. However, there are other methods that could have been used and that achieve a better performance on datasets that are perhaps more similar to the setting of dataset retrieval (e.g., methods with better performance on benchmarks such as BEIR). The out-of-domain experiments should ideally also consider models such as those on the top results for the Massive Text Embedding Benchmark ( see https://huggingface.co/spaces/mteb/leaderboard ).

B - On what regards score interpolation, instead of using the arithmetic mean of normalized scores, the authors should instead leverage methods from the information retrieval literature on rank fusion (e.g., methods such as reciprocal rank fusion -- see for instance https://amenra.github.io/ranx/fusion/ for an implementation of several of these methods ).

**Paper Topic And Main Contributions:**

The authors present a comparative study on different methods for the task of ad-hoc dataset retrieval, which concerns retrieving datasets matching a query with a textual description, from a large collection of datasets where the instances are described through a set of English metadata elements (e.g., title, description, author, or tags). Specifically, using two datasets from previous work (ACORDAR and NTCIR-E), the authors evaluated a field-weighted BM25 baseline against dense retrieval approaches (using models trained on MS-MARCO, or further fine-tuned with in-domain data), against approaches based on entities extracted through off-the-shelf entity linking methods, or against approaches that combine dense retrieval and entities. Experimental results show that combined methods can achieve a better performance.

**Questions For The Authors:**

A - In Section 3.3.2, when first discussing the use of the harmonic mean to combine two results, the authors should briefly describe what are some of the properties of this approach that make it preferable to the arithmetic mean (e.g., the fact that having a high score with the harmonic mean requires both input scores to also have high values, and not just one of them).

B - In Section 3.2.1, the authors state that "entity linking is an established research problem and we use off-the-shelf tools in the experiments." However, the performance of entity linking systems often degrades on short texts (such as queries), given that there is not enough context to disambiguate entities. Ideally, the paper should discuss the accuracy/performance of the entity linking methods on the data in which they are being applied.

C - Section 3.2.3 should ideally describe in more detail the embeddings for the entities that are used in the computation of entity set similarity and the entity-level similarity. Later in Section 4.3.2 the authors state that they use either Wikipedia2vec or KGTK for the entity set similarity, but they also say that Wikipedia2vec was also used as the source of word embeddings for entity-level similarity (which does not seem to make much sense, since I believe that only the words that correspond to a Wikipedia concept will be included, rather that the words that are part of entities). Why haven't the authors considered the use of more standard word embedding models?

D - The description of the neural retrieval models used in the experiments should provide more details. It should be noted that some of these models correspond to cross-encoders (which only make sense to use in a re-ranking setting), whereas some of the models correspond to dual encoders (which would make more sense to use as first-stage retrievers). The different models are also associated to different training strategies and loss functions. For instance, the main innovation in ANCE is in the way the negative samples are obtained for training, but the authors seem to be using always the same training procedure when fine-tuning the models to the in-domain data (and aspects of this training procedure, such as the loss function, are not explained in the paper). On were in fact these models trained with the in-domain data?

**Reasons To Accept:**

* The paper is clear and well-written, addressing an interesting topic that has not received much attention.

* The extensive experimental results that are reported on the paper will likely constitute a useful benchmark for people working on dataset retrieval.

**Reasons To Reject:**

* The re-ranking setup is not the best evaluation setup that could have been considered. From the description provided in Section 4.2, the authors seem to be evaluating the different methods on a re-ranking setup, using the different models to re-rank the top-10 results provided by field-weighted BM25. However, the re-ranking of only the top-10 results is too limiting, and there are few opportunities to see increased effectiveness when focusing only on this small list. Ideally, the authors should report on experiments focused on re-ranking a larger set of datasets returned by first-stage retrieval.

* The paper does not give much details about some of the models/methods that are being used (e.g., it does not describe the training of the neural retrieval models with in-domain data in sufficient detail, and it does not address the performance of the pre-existing entity linking systems on the in-domain data).

* A different approach should have been considered for combining the different retrieval methods, more in alignment with previous rank fusion methods in the information retrieval literature.

**Reproducibility:**

4: Could mostly reproduce the results, but there may be some variation because of sample variance or minor variations in their interpretation of the protocol or method.

**Reviewer Confidence:**

4: Quite sure. I tried to check the important points carefully. It's unlikely, though conceivable, that I missed something that should affect my ratings.

---

> ### Author Rebuttal · Authors · 2023-08-29
>
> Many thanks for the time and detailed comments from the reviewer.
>
> Firstly, we would like to clarify that the reviewer's **reasons to reject are mainly sourced from the limited availability of test collections rather than from our experimental design**.
>
> Specifically, the first reason to reject is "the re-ranking of only the top-10 results is too limiting". We clarify that **the available test collections such as ACORDAR only provide annotated relevance labels for top-10 retrieval results**. Therefore, although we agree that it would be interesting to re-rank "a larger set of datasets returned by first-stage retrieval" as suggested by the reviewer, it is not feasible to conduct such an experiment based on existing test collections.
>
> The second reason to reject (related to **Question B**) is "it does not address the performance of the pre-existing entity linking systems on the in-domain data". Again, it is difficult to conduct such an experiment because **the available test collections for ad hoc dataset retrieval do not provide gold-standard entity annotations**. However, we agree that such an analysis would be helpful, and we plan to extend existing test collections with our own annotations, but it has to be in our future work.
>
> This reason to reject is also related to **Question D**: details about the training of the neural retrieval models with in-domain data. The reviewer said "the authors seem to be using always the same training procedure". **This is a misunderstanding.** We clarify that we directly used their original source code and we followed **their own training strategies and loss functions**. For example, for a fair comparison, we fed the same training data into all the models which were treated as black boxes, but if a model such as ANCE had a special negative sampling strategy inside the black box, it would certainly be executed. We will add this clarification to the paper.
>
> We appreciate the reviewer's **suggestion in Missing References** about adding experiments using other rank fusion methods. **We have completed this experiment, which also helps address the reviewer's last reason to reject.** Specifically, by using the suggested ranx library, we tried different fusion methods: "min", "max", "med", "sum", "anz", "mnz", "isr", "log_isr, "bordafuse", and "condorcet". We observed that "sum" (which is equivalent to our arithmetic mean) and "mnz" are the two best-performing methods, which empirically supported our choice of arithmetic mean in the paper. We will add this experiment to the appendix.
>
> Next, we answer the remaining questions in the review.
>
> **Question A is a minor suggestion on writing**, which we will easily incorporate into the camera-ready version.
>
> Regarding **Question C**, the reviewer said "I believe that only the words that correspond to a Wikipedia concept will be included". **This comment is inaccurate.** Wikipedia2vec provides embeddings for all the words occurring in Wikipedia articles, not limited to those in Wikipedia concepts/titles; that is, Wikipedia2vec is exactly one of the "standard word embedding models" suggested by the reviewer.

---

### Official Review · Reviewer_RyqX · 2023-08-04

**Soundness:** 3

**Excitement:**

2: Mediocre: This paper makes marginal contributions (vs non-contemporaneous work), so I would rather not see it in the conference.

**Paper Topic And Main Contributions:**

An analysis of the different approaches for ad-hoc dataset retrieval. The authors carry out an extensive study on what are the methods that work best.


**Reasons To Accept:**

- The study on ad-hoc dataset retrieval is thorough and captures a wide variety of approaches on a diverse set of corpora.

**Reasons To Reject:**

- The paper does not present any novel findings, either at problem setting, methodology, etc. Its main contributions are on surveying already existing approaches, which in my opinion it is not enough for this conference.

- The scope of this problem is marginally relevant for the EMNLP conference.

**Reproducibility:**

3: Could reproduce the results with some difficulty. The settings of parameters are underspecified or subjectively determined; the training/evaluation data are not widely available.

**Reviewer Confidence:**

4: Quite sure. I tried to check the important points carefully. It's unlikely, though conceivable, that I missed something that should affect my ratings.

---

> ### Author Rebuttal · Authors · 2023-08-27
>
> Many thanks for the reviewer's time. However, **we could not agree with the reviewer's two very general reasons to reject**.
>
> R1. The reviewer said "the paper does not present any novel findings" and "its main contributions are on surveying already existing approaches".
>
> **Both comments are inaccurate.**
>
> Our main contributions are **not a survey but an extensive empirical investigation**, and we are among the first to conduct such an investigation for the task of ad hoc dataset retrieval which is a relatively new and under-explored IR task. Our experiments provided **a lot of new and insightful findings** (see Section 5 for details and Section 6 for a summary), which successfully answered three Research Questions posed in the Introduction section.
>
> R2. The reviewer said "this problem is marginally relevant for the EMNLP conference".
>
> **We could not agree because the relevance is explicit.** Ad hoc dataset retrieval is an emerging IR task, which is clearly within the scope of the Information Retrieval and Text Mining track.

---

### Official Review · Reviewer_SVC3 · 2023-08-08

**Soundness:** 2

**Excitement:**

2: Mediocre: This paper makes marginal contributions (vs non-contemporaneous work), so I would rather not see it in the conference.

**Paper Topic And Main Contributions:**

This paper investigates the utility of Large Language Models (which the authors call "implicit knowledge") and knowledge graphs (which the authors call "explicit knowledge") for the task of dataset retrieval. The main contribution is that the authors find out that their proposed methods based on LLMs and knowledge graphs are ineffective when used on their own, but can be effective when interpolated with BM25 scores. The latter claim is, however, supported only by some experimental results.

**Questions For The Authors:**

* why didn't you experiment with other functions (geometric mean, maximum, top-k) besides arithmetic mean for aggregating row and column maximum values?

* why do you interpolate the proposed methods with BM25 by just calculating the arithmetic mean of the two scores and not as a weighted interpolation, which would give you an opportunity to find the optimal contribution from both methods?

**Reasons To Accept:**

* This paper appears to be the first study to examine the utility of LLMs and knowledge graphs for dataset retrieval

**Reasons To Reject:**

* Lack of experimentation and analysis: the proposed method for scoring the similarity of document and query, which is the only novelty of this paper from the technical perspective, is non-parametric. However, there are many options and alternatives within the proposed method that should have been explored to better understand its functioning and capabilities:

It would be interesting to find out what are the relative contributions of semantic vs. lexical similarity functions in computing entity similarity

It would be interesting to see if other functions (geometric mean, maximum, top-k) besides arithmetic mean for aggregating row and column maximum values would be more effective

It would be interesting to experiment with a parametric similarity function (e_i W e_j, where W is a parameter) besides cosine similarity

It seems sub-optimal to interpolate the proposed methods with BM25 by just calculating the arithmetic mean of the two scores. In prior IR research, such interpolation is normally done as a linear combination.

* Lack of illustrative examples: it is not at all clear why measuring similarity between a query and dataset metadata in terms of entities would benefit retrieval. A clear and convincing illustrative example would make the proposed approach stronger

* Unconvincing experimental results: the results presented in Tables 3, 4, 5 and 6 only partially support the proposed ideas. Specifically:

- Table 3: the proposed explicit knowledge-enhanced methods result in statistically significant improvement of *only one metric* in *only one setting* across both datasets
- Table 4: the claim that the proposed implicit knowledge-enhanced methods interpolated with BM25 are effective is supported by statistically significant improvement of *2 out of 4 metrics only in the case of 1 out of 5 LLMs* on the NTCIR-E dataset
- Table 5: the claim that the proposed explicit knowledge-enhanced methods interpolated with BM25 are effective is not supported by statistically significant improvement *in any setting* on the NTCIR-E dataset
- Table 6: the claim that the proposed implicit and explicit knowledge-enhanced methods complement each other when interpolated with BM25 is not supported by any statistically significant improvement on the NTCIR-E dataset.

Overall, the paper leaves a feeling that the authors were trying to jam-pack two ideas (utilizing LLMs and knowledge graphs in dataset retrieval) at the expense of properly exploring and determining the true potential of the most promising one - utilizing knowledge graphs in dataset retrieval.  I would suggest the authors focus on just knowledge graphs and properly explore the potential of this idea through more extensive experiments.

**Reproducibility:**

3: Could reproduce the results with some difficulty. The settings of parameters are underspecified or subjectively determined; the training/evaluation data are not widely available.

**Reviewer Confidence:**

4: Quite sure. I tried to check the important points carefully. It's unlikely, though conceivable, that I missed something that should affect my ratings.

---

> ### Author Rebuttal · Authors · 2023-08-28
>
> Many thanks for the time and comments from the reviewer.
>
> Firstly, we answer the reviewer's **questions** in the review, which also helps **address the reviewer's first reason to reject**.
>
> Q1. why didn't you experiment with other functions (geometric mean, maximum, top-k) besides arithmetic mean for aggregating row and column maximum values?
>
> **Actually we have experimented with other functions.** For example, below Equation (7) we have explained why we empirically chose harmonic mean. Similarly, for Equations (3) and (4), we chose arithmetic mean because it outperformed other functions in our experiments. **We did not include these experimental results in the paper because they may distractingly overload readers who might overlook our main results.** Since the reviewer referred to these results, we will add them to the appendix.
>
> Q2. why do you interpolate the proposed methods with BM25 by just calculating the arithmetic mean of the two scores and not as a weighted interpolation, which would give you an opportunity to find the optimal contribution from both methods?
>
> **We have tried a weighted interpolation.** However, due to the small training sets, we could not learn an effective weight, and the result on the test set was even worse than that by simply taking the arithmetic mean. We will add this explanation to the paper, and add the detailed results to the appendix.
>
> The reviewer's **second reason to reject** is related to **presentation issues** and can be easily addressed in the camera-ready version.
>
> Next, we believe that **the reviewer's third reason to reject is debatable**. First, the reviewer said "no statistically significant improvement after interpolation with BM25 for any LLM". **This comment is inaccurate.** As shown in Table 4, we observed statistically significant improvements with ColBERT. Second, please note that we observed **statistically significant improvements in most of (though not all) the settings**, and **in the remaining settings the improvements are often considerable** (e.g., +0.02 or +0.03) although not significant under p<0.05. We believe that such results are sufficient for supporting the observations we summarized, but we respect the reviewer's comment and are willing to further revise our expressions if required.

---

### Official Review · Reviewer_Pb56 · 2023-08-11

**Soundness:** 4

**Excitement:**

3: Ambivalent: It has merits (e.g., it reports state-of-the-art results, the idea is nice), but there are key weaknesses (e.g., it describes incremental work), and it can significantly benefit from another round of revision. However, I won't object to accepting it if my co-reviewers champion it.

**Paper Topic And Main Contributions:**

This work explores the effectiveness of incorporating knowledge into the emerging task of ad hoc dataset retrieval, where the authors conducted a systematic investigation of implicit and explicit knowledge-enhanced methods for ad hoc dataset retrieval on two test collections.


**Questions For The Authors:**

1. In subsection 5.1, why coCondenser, ColBERT, and ANCE achieve more significant improvements than monoBERT and monoT5 in in-domain setting? Also, why monoT5, coCondenser, and ANCE achieve more significant improvements than Col-BERT in out-domain setting?

**Reasons To Accept:**

1. The article is easy to follow.
2. The article provides a good literature survey of the field.
3. The authors show the effectiveness of knowledge aware retrieval technique on dataset retrieval task and reveal the strengths and weaknesses of employed methods, which give some successful cases to this underexplored task.


**Reasons To Reject:**

1. Although this work explore more effective methods on dataset retrieval task, the employed dense retrieval techniques are well-established in information retrieval-related tasks which does not bring much inspiration.
2. The analysis on the experimental results is not strong, more detailed insights explaining the observations are needed.


**Reproducibility:**

4: Could mostly reproduce the results, but there may be some variation because of sample variance or minor variations in their interpretation of the protocol or method.

**Reviewer Confidence:**

4: Quite sure. I tried to check the important points carefully. It's unlikely, though conceivable, that I missed something that should affect my ratings.

---

> ### Author Rebuttal · Authors · 2023-08-28
>
> Many thanks for the time and comments from the reviewer.
>
> Firstly, we answer the reviewer's **question**, which also **addresses the reviewer's second reason to reject**.
>
> Question: Why coCondenser, ColBERT, and ANCE achieve more significant improvements than monoBERT and monoT5 in in-domain setting? Also, why monoT5, coCondenser, and ANCE achieve more significant improvements than Col-BERT in out-domain setting?
>
> Our answer:
>
> In an in-domain setting, as mentioned in Section 5.1, since the training sets are small, models might have overfitted. For monoBERT and monoT5, these models perform extensive query-document interaction, probably leading to **more severe overfitting** and worse results on the test set.
>
> In an out-of-domain setting, ColBERT which relies on word-level matching achieved relatively bad results possibly due to some ambiguous query words such as 'MCI' which could be mistakenly matched with the same word having a different meaning in an irrelevant document. Other models relying on the global [CLS] token representation were **less influenced by such individual word mismatches**.
>
> We will incorporate the above **insights** into the camera-ready version.
>
> Next, we would like to **clarify about the reviewer's first reason to reject** about novelty. Please note that our work addresses ad hoc dataset retrieval which is a relatively new and under-explored IR task. The methods we applied are "well-established" in conventional IR tasks but **their effectiveness in this emerging task remains unknown**, as characterized by our three research questions posed in the Introduction section. Our work **fills this gap** by conducting an extensive empirical investigation to answer these research questions and to help establish the current best practice for this task.

---

### Meta-Review · Area_Chair_48Tx · 2023-09-14

**Recommendation:** 4

**Metareview:**

The paper investigates the utility of incorporating knowledge into Large Language Models for the task of ad hoc dataset retrieval. The authors conducted a systematic investigation of implicit and explicit knowledge-enhanced methods for ad hoc dataset retrieval on two test collections. The main contribution is that the authors evaluated a field-weighted BM25 baseline against dense retrieval approaches against approaches based on entities extracted using established entity linking methods and against approaches that combine dense retrieval and entities. Experimental results show that combined methods can achieve a better performance.

The approach presented is sound and the paper is clear and well written. The arguments are well supported by results and analysis.

---

### Decision · Program_Chairs · 2023-10-07

**Decision:**

Accept-Findings

**Comment:**

The paper investigates the utility of incorporating knowledge into Large Language Models for the task of ad hoc dataset retrieval. The authors conducted a systematic investigation of implicit and explicit knowledge-enhanced methods for ad hoc dataset retrieval on two test collections. The main contribution is that the authors evaluated a field-weighted BM25 baseline against dense retrieval approaches against approaches based on entities extracted using established entity linking methods and against approaches that combine dense retrieval and entities. Experimental results show that combined methods can achieve a better performance.

The approach presented is sound and the paper is clear and well written. The arguments are well supported by results and analysis.